# A Bayesian Approach to Adversarially Robust Life Testing

Dorina Weichert [1]   Sebastian Houben [2]   Alexander Kister [3]   Gunar Ernis [1]   Tim Wirtz [1]

## Abstract

In materials science and engineering, the lifetime of materials and products is tested by costly manual characterization procedures that are standardized only in certain cases. In this paper, we investigate a modular Bayesian approach to lifetime testing that can reduce the number of experiments and, thus, the overall cost of experiments. The approach is based on the correct definition of the probability of the outcome of an experiment, e.g., its likelihood. Since this is usually unknown, we extend it to the adversarial setting, finding an experimental procedure that is robust to a given set of probabilities in the worst case. By simulations, we empirically show the advantages of this procedure over the state-of-the-art and the basic approach, potentially reducing the number of costly experiments.

## 1. Introduction

Life testing describes the planning, execution, and analysis of product tests to estimate a product's expected lifetime, e.g., in engineering or material science.

To estimate a product's lifetime, first, the most critical factors stressing the product are identified. In practice, these typically refer to alternating stresses, e.g., alternating mechanical loads, alternating temperatures, or alternating electrical loads. Then, so-called accelerated tests are taken out, where the identified stresses are cyclically applied to a sample in short intervals to determine the maximum stress a sample can withstand. We generate a test statistic to find this maximum stress: we apply different stress levels to different samples and record if they break or not - a so-called accelerated binary test (Escobar & Meeker, 2006, p. 4).

In the ideal setting, if the sample survives a predefined number of cycles that are thought to approximate an infinite lifetime (a so-called *survivor*), we infer that the product is resistant to this load. If it fails the test (a so-called *failure*), we infer that the product does not withstand this load for its whole lifetime. Unfortunately, the real setting is noisy. The product quality and, therefore, the samples vary, so at some stress levels, some samples fail, and some survive the testing procedure. Therefore, practical life tests aim to either find the stress level that refers to a specific failure probability or the estimation of a function expression for the failure probability depending on the load.

There is no general standard procedure for life testing, but standards exist for special applications, e.g., fatigue testing of steels (DIN 50100:2016-12, 2016). Also, the analysis of life test results is not standardized, but a spectrum of heuristics, likelihood-based methods, and Bayesian approaches exists. Please see the comprehensive standard work by Meeker et al. (2022) for an overview.

These approaches (the testing and analysis procedures) have multiple drawbacks. On the one hand, they do not explicitly consider the similarity of different products. On the other hand, the likelihood-based analysis procedure is sensitive against false assumptions, e.g., the assumption of the false failure probability distribution.

Our work uses Bayesian methods to find a testing procedure and analysis approach that explicitly considers the similarity of different products, is highly efficient, and worst-case robust against possible misspecifications. As shown in the motivating figure 1, it consists of two connected modules. In the first module, we use a Machine Learning model to implement the similarity of different products based on historical data and expert knowledge. When now testing a new product, we use the prediction of this model as a prior for a traditional Bayesian Inference setup that offers the opportunity to a) create a Maximum a Posteriori estimate of failure behavior related quantities, to b) estimate standard deviations of the parameters to be used as a confidence estimate, and to c) derive a robust acquisition function, serving as a testing protocol.

In an application study from fatigue testing, we show the suitability of our approach for general life testing scenarios.

[1] Fraunhofer Institute for Intelligent Analysis and Information Systems IAIS, Sankt Augustin, Germany [2] University of Applied Sciences Bonn-Rhein-Sieg, Sankt Augustin, Germany [3] VP.1 eScience, Federal Institute for Materials Research and Testing BAM, Berlin, Germany. Correspondence to: Dorina Weichert <dorina.weichert@iais.fraunhofer.de>.

*Accepted at the 1st Machine Learning for Life and Material Sciences Workshop at ICML 2024.* Copyright 2024 by the author(s).

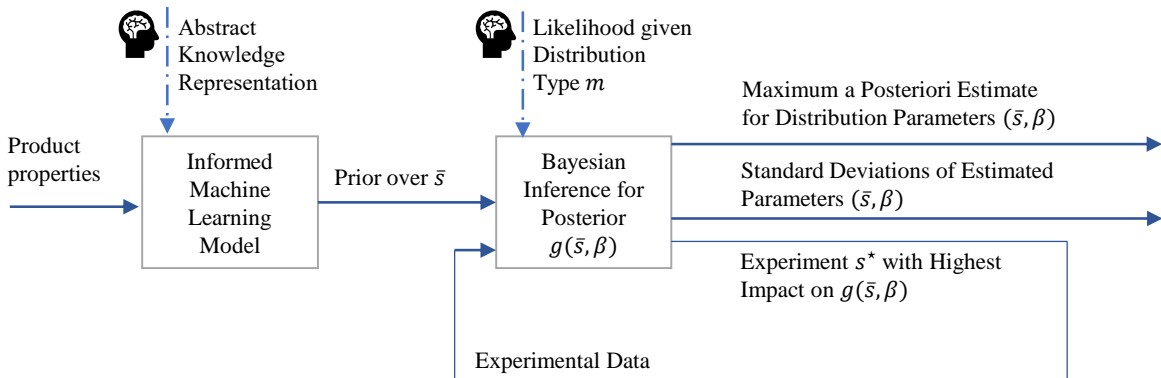

*Figure 1.* Bayesian approach for life testing. While the similarity of materials is captured in the Machine Learning model in the first module, the second one allows for estimation of relevant quantities via traditional Bayesian Inference.

## 2. Background

In the following, we first give an overview of the related work and afterward introduce the staircase approach. This approach is a method designed for fatigue testing which is transferable to general life testing problems.

### 2.1. Related Work

For brevity, we narrow down the related work to Bayesian approaches for data analysis and data acquisition that are suitable for accelerated binary tests and highlight the most relevant ones for our approach. For other types of tests or non-Bayesian approaches, we like to refer the reader to the standard work by Meeker et al. (2022).

Bayesian analysis of reliability data has a long tradition, see e.g., chapter 10 of Meeker et al. (2022) for an overview. To estimate the unknown parameters of the failure probability distribution, authors make use of conjugate priors (Barnett, 1972), other analytical approximations (Soman & Misra, 1994), and of Markov Chain Monte Carlo (MCMC) methods, see e.g., Zhang & Meeker (2006); Li & Meeker (2014); Shuto & Amemiya (2022). In our approach, we use a grid-based sampling approach, significantly speeding up the approximation of the posterior in contrast to the use of MCMC methods, even though slightly downgrading precision. As the quality of the Bayesian analysis highly depends on the choice of the prior, several authors, e.g., Tian et al. (2024); Li & Meeker (2014) study the impact of the prior explicitly. As expected, they find that the quality of the posterior estimates depends on the choice of prior: if it is too narrow and far from the true values, it hinders finding the true estimates, but a reasonable prior supports the estimation of the unknown parameters.

The idea to use the Bayesian framework also for data acquisition is newer (Insua et al., 2020; Limon et al., 2017). The most similar acquisition functions to ours are the use of the variance of the predictive posterior (Yili Hong & Meeker, 2015), the Kullback-Leibler-divergence of the actual estimate and the predictive posterior (Xu & Tang, 2015). The latter approach, as well as ours that uses the entropy of the predictive posterior distribution, are very useful in a potential multi-modal setup that the variance approach does not cover.

Even though the approaches above are described for arbitrary likelihoods, experiments mainly focus on the use of special types, such as the Weibull distribution (Yili Hong & Meeker, 2015; Zhang & Meeker, 2006; Xu & Tang, 2015) or the log-normal distribution (Yili Hong & Meeker, 2015). Even though we are working in the same Bayesian setup, we twist the acquisition function such that it tackles the fact that the underlying ground truth is unknown, treating the special case of an adversarial choice of distribution.

### 2.2. Staircase Approach for Fatigue Testing

Unfortunately, there is no general approach to life testing. This section describes the so-called staircase approach for fatigue testing, as defined in DIN 50100:2016-12 (2016, pp. 45-50, pp. 56-59). It is transferable to life tests of other products. These are often also carried out in a staircase procedure, but this is not clearly defined (Meeker et al., 2022, pp. 289-309).

As we use the staircase method for later benchmarking our approach, we quickly revisit its working principle. Before applying the staircase method, the process engineer defines load levels for experimentation. These load levels are defined by $L_i = L_{ini} \cdot d^i$, where $L_{ini}$ is a user-defined initial

load, $d$ is the user-defined step size, and $i \in \mathbb{Z}$.

Given the load levels, experimentation follows a strict protocol: if a specimen at a specific load is a failure, the load level for the next experiment is reduced by one step; if it is a runout, the load level is raised by one step.

For later analysis, the generated experimental series has to fulfill multiple requirements: in a valid series, the initial load level is reached at least once again during experimentation. Additionally, the series must contain at least three load levels and at least two turning points where a runout is followed by a failure or the other way around.

Given a valid experimental series, the mean fatigue strength is estimated using two parameters: the lowest valid load level $L_0$ and the number $l_k$ each load level was reached, where $k \in \mathbb{N}_0$ and $k = 0$ refers to the lowest valid load level.

Generally, the staircase method assumes the failure probability of a sample to follow the cumulative density function of a log-normal distribution with mean $\mu_L$ and standard deviation $\sigma_L$. The mean $\mu_L$ is then found by:

$$\mu_L = L_0 \cdot \frac{\sum_k k \cdot l_k}{\sum_k l_k} \ . \qquad (1)$$

The standard deviation $\sigma_L$ is calculated using a more complex heuristic (DIN 50100:2016-12, 2016, p. 58). In practice, the reliable estimation of this value requires many experiments and is rarely performed.

Looking at the staircase method, potential disadvantages become clear: its efficiency depends heavily on the experience of the test engineer, who determines the initial value of the method and the step size. These values also include the similarity of one product with others. In addition, the assumption of a log-normal distribution can be disadvantageous if the ground truth is different.

## 3. Bayesian Life Testing

The main requirement for life testing approaches is to obtain a sufficiently precise estimate of the stress-dependent failure behavior with a high sample efficiency. This efficiency is motivated by the high induced test costs, e.g., a single fatigue test can cost up to $\$10,000$.

To face this requirement, we include as much data and expert knowledge as possible in our approach. Before giving a detailed description, let us briefly summarize our basic assumptions.

### 3.1. Basic Assumptions

First of all, the probability $p_{\text{failure}}$ of a sample to fail the life test at a particular stress follows a monotonically increasing function $\Phi(s) : \mathcal{S} \mapsto [0, 1], s \in \mathbb{R}$. In turn, we can also calculate the survival probability of a sample $p_{\text{survivor}}$ by $\overline{\Phi(s)} = 1 - \Phi(s)$. Typically, this function is expressed by the cumulative density function of a parametric heavy-tailed probability distribution $m$, such as the log-normal, the Weibull, or the Gumbel distribution (Meeker et al., 2022, pp. 66-90). For the sake of simplicity, we will concentrate on two-parametric distributions with a location parameter $\bar{s}$ and a scale parameter $\beta$ in the following.

We are not able to observe any of these functions directly, but if applying a certain stress level, we observe if the probe either fails or survives the test. Given a statistic over failures, indexed by $i$, and survivors, indexed by $j$, we are able to express the likelihood of the unknown parameters of $\Phi_m(s)$ by

$$e_m(\bar{s}, \beta) = \prod_i \Phi_{m;\bar{s},\beta}(s_i) \cdot \prod_j (1 - \Phi_{m;\bar{s},\beta}(s_j)) \ . \qquad (2)$$

Unfortunately, most open-source data on life tests is only available in aggregated form, i.e., in terms $\bar{s}$, instead of the overall test statistics. Examples include, e.g., the open fatigue data sheet[1], or the open space use materials strength data sheet[2]. It usually corresponds to a specific quantile of the resulting distribution, e.g., the stress where the probability of failure corresponds to 90 %. In our setup, we assume that $\Phi(s)$ can be expressed in terms of the maximum stress as the location parameter $\bar{s}$.

### 3.2. Method

To include as much information as possible, we conduct a two-modular approach. We capture the general product properties in the first module and learn a Machine Learning model for the location parameter $\bar{s}$. In the second module, we describe the expectations about the individual product's response to stress. We connect the modules by using the prediction from the first module as prior for the second one, so we combine the available data and knowledge to obtain a highly efficient approach.

MODULE 1: MACHINE LEARNING MODEL

In the first module, we apply a Machine Learning model that is able to predict a distribution over the location parameter $p(\bar{s})$. In our application study, we opt for a Gaussian Process model with an engineered covariance function to include assumptions on the material behavior by domain experts, but other models, such as Bayesian linear regression or a Random Forest, are also suitable.

---

[1]https://fds.nims.go.jp/
[2]https://sds.nims.go.jp/

MODULE 2: BAYESIAN INFERENCE

Given the prior distribution on the location parameter $p(\bar{s})$ from the first module and the expression for the likelihood in equation (2), we find the posterior distribution over the unknown parameters of $p_{\text{failure}}$ by

$$g(\bar{s}, \beta) = p(\bar{s}) \cdot p(\beta) \cdot e_m(\bar{s}, \beta) . \qquad (3)$$

For $p(\beta)$, any prior is possible as long as it is valid for the assumed underlying distribution, e.g., positive for the Weibull distribution.

Given the equation for the posterior distribution, we can find helpful quantities for the daily work of life-testing engineers, namely, a Maximum a Posteriori estimate, a confidence estimate, and an acquisition function.

**Maximum a Posteriori Estimate**  The Maximum a Posteriori estimate is defined as the argmax location of the posterior and thus corresponds to the most probable parameters. We find

$$\hat{\bar{s}}, \hat{\beta} = \arg\max_{\bar{s}, \beta} g_m(\bar{s}, \beta) . \qquad (4)$$

Often, estimates $\hat{\bar{s}}$ from historical measurements is used as label data of module 1.

**Estimation of Confidence**  To provide information about the quality of the Maximum a Posteriori estimate, we vote for using the standard deviation of the joint posterior, marginalized in the direction of each parameter, $\text{Std}(g_m(\bar{s}, \beta))$. This estimation is risky to potential misinterpretation if the priors $p(\beta)$ or $p(\bar{s})$ are wrong or too tight but very valuable in the well-conditioned case.

**Definition of Acquisition Function**  Given the posterior distribution $g(\bar{s}, \beta)$, we are also able to approximate its entropy $H$ via sampling. The entropy is an essential measure of uncertainty and is used in information-based acquisition functions, which are well-known for their high efficiency, for example, in Bayesian Optimization (Garnett, 2023, p. 136), Bayesian Optimal Experimental Design (Lindley, 1956), or Active Learning (Settles, 2012, pp. 13-16).

However, we cannot only approximate the entropy given the actual dataset, but we can also estimate the effect of adding an experiment at stress $s$ at timestep $t$. As the experiment either fails or survives, we must approximate the resulting entropy for both cases. Additionally, we weigh these entropies by the probability of the considered outcomes using the Maximum a Posteriori estimates $\hat{\bar{s}}_t, \hat{\beta}_t$ at timestep $t$.

Given a specific model $m$, we find:

$$\begin{aligned}
\alpha_m(s) = \\
(-H(g_m(\bar{s}, \beta | \text{outcome}(s) = \text{failure}, s))) \\
\cdot \Phi_{m_{\hat{\bar{s}}, \hat{\beta}}}(s) \qquad (5) \\
+ (-H(g_m(\bar{s}, \beta | \text{outcome}(s) = \text{survivor}, s))) \\
\cdot (1 - \Phi_{m_{\hat{\bar{s}}, \hat{\beta}}}(s)) .
\end{aligned}$$

The next experiment is taken out at $\arg\max_{s \in \mathcal{S}} \alpha_m(s)$.

### 3.3. Adversarially Robust Adaption

We extend our formulation of the acquisition function in equation (5) to the setting, where the underlying model $m$ of the failure probability $p_{\text{failure}}$ is unknown, which is the typical case (Meeker et al., 2022, pp. 17-19).

In our adaption, we take a conservative approach and consider the adversarial setting. This approach is highly relevant in engineering applications, where estimating a system's behavior (in our case, the failure behavior) under worst-case assumptions is essential. For our acquisition function, this means that we aim to add the experiment that provides maximum information about the distribution parameters of the model $m$ with the highest uncertainty. Given a set $\mathcal{M}$ of potential models, we find

$$s^{\star} = \arg\max_{s \in \mathcal{S}} \min_{m \in \mathcal{M}} \alpha_m(s) . \qquad (6)$$

## 4. Case Study: Adversarially Robust Fatigue Strength Estimation

The following case study shows the application of our approach for the fatigue strength estimation of stainless steels. It is transferable to other settings of accelerated binary testing. Fatigue strength is "the value of stress at which failure occurs after $N_f$ cycles" (ASTM E1823-24a, 2024), in our case $N_f = 10^7$. Our example examines the fatigue strength against a switching tensile and compressive load with zero mean stress. The median fatigue strength $\bar{s}$ is the estimate, where the failure probability is 50 % (DIN 50100:2016-12, 2016, p. 5). Knowing the fatigue strength of a material is of high practical importance: the measure is used in the design of products such as gears, springs, or other highly stressed mechanical components.

### 4.1. Machine Learning Model

In the first module, we create Gaussian Process  with a tailored covariance function to include existing historical data and expert knowledge. The training data was offered by a partner company and is based on fatigue data from 114 stainless steels. Unfortunately, we are not allowed to publish the data and thus the final trained model, as the GP offers

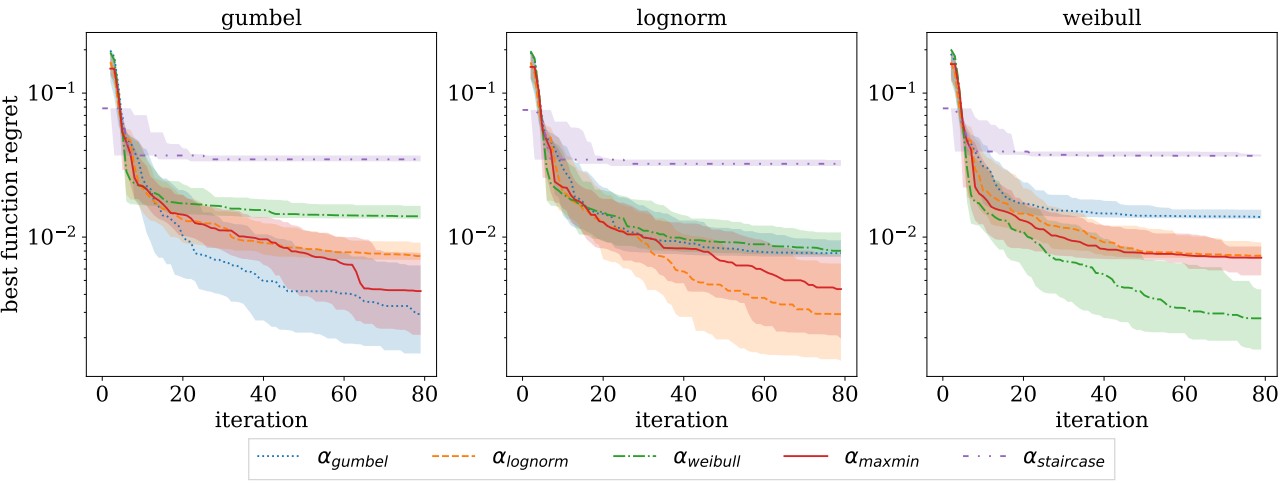

*Figure 2.* Best function regret over iteration. We report the median and the quantiles of 100 runs. The robust maxmin acquisition function from equation (6) performs worse than when assuming the actual (unknown) ground truth, but better than when using a wrong assumption or the heuristic staircase method.

direct access to the train data. Our model uses four relevant dimensions of an experiment as input: the loaded volume V90 of the specimen, the specimen's edge hardness, the load type (e.g., bending, stress, and strain), and the load ratio $R$, which describes the ratio between the maximum and the minimum load amplitude. The output for training was the related median fatigue strength $\bar{s}$ estimated via historical experiments.

Before training the model, we make a train-test split of size 80/20. To cope with the data's non-normality, we logarithmize the mean fatigue strength data before applying standardization. Afterward, we use a constant zero mean function and a tailored covariance function $k_L$ that was defined with the help of process engineers. This covariance function merges the assumption of a linear trend with an expected very smooth local behavior by summing a linear covariance function $k_{lin}$ with automatic relevance determination with a rational quadratic covariance function $k_{RQ}$. Thus, our covariance function is defined as follows:

$$k_L(x, x') = \sum_{d=1}^{D} \sigma_d^2 x_d x_d' + \left(1 + \frac{(x - x')^2}{2\alpha\sigma_l^2}\right)^{-\alpha} .$$

Here, $x$ are our input parameters, while $\sigma_d$, $\sigma_l$, and $\alpha$ are hyperparameters of the covariance function that can be estimated by maximizing the marginal log-likelihood. To validate the approach, we perform a 10-fold cross-validation, also comparing alternative covariance functions (a radial basis function covariance function and a Matérn class covariance function in sum and product combination with a linear covariance function), each time estimating the covariance

function's hyperparameters using maximum marginal log-likelihood. For testing purposes, we condition the model with the best performance (selecting the covariance function and its hyperparameters from the best fold) on all train data and find a model performance of $R^2 = 0.91$, which is comparable to a state-of-the-art model (Agrawal et al., 2014, p. 13). For later use as a prior in the Bayesian Inference module, we condition the model on all available data afterward, keeping the hyperparameters fixed.

### 4.2. Studying Acquisition Functions

To study the behavior of the acquisition functions, we simulate an experiment for a stainless steel type with failure probability $\Phi^\star$ with median fatigue strength of $\bar{s}^\star = 400$ N, which refers to steel C15 (1.0401), a non-alloy quality steel. We assume the failure probability distribution to have a standard deviation of $10^{0.4}$ N.

We use the following ground truth models $m \in \mathcal{M}$: the cumulative distribution functions of the Gumbel, of the lognormal and of the Weibull distribution, which we rewrite to depend on the median $\bar{s}$ as the location parameter and another free parameter $\beta$. For each distribution, we calibrate $\beta$ to match the expected standard deviation. Please see the supplementary material, section C for a visualization of the failure models.

We virtually run the staircase acquisition function, the non-robust approaches following equation (5), and the robust approach from equation (6) for $n = 80$ iterations and 100 repetitions. We use positive normal priors for each distribution's $\bar{s}$ and $\beta$, $\bar{s}$ is $\mathcal{N}(\mu = 400, \sigma^2 = 100^2)$ N, while the

prior over $\beta$ is calibrated such that the entropy of $g(\bar{s}, \beta)$ is approximately the same after observing a survivor at $300$ N and a failure at $500$ N. We do not use the prediction of our model as it is very close to the ground truth with a small standard deviation, and we want to enable a fair comparison with the staircase method that only indirectly takes into account the similarity with other materials. To approximate the posterior distribution via sampling, we use $10^5$ samples from each of the priors.

We report the best function regret, which we define as $\min_{t\in[1,n]} |\Phi^\star - \Phi_{m_{\hat{s},\hat{\beta}},t}|$, where $\Phi_{m_{\hat{s},\hat{\beta}},t}$ is the distribution defined by the Maximum a Posteriori parameter estimates at iteration $t$.

As expected, the best approximations are made when ground truth and approximating distributions are the same. The robust acquisition function is performing second-best, so it shows that it is indeed helpful to characterize the failure probability for cases where the ground truth is unknown. While the staircase method is comparable to the other acquisition functions in the first iterations, it seems to diverge for a higher number of samples. This is due to the fact that it struggles with determining the correct standard deviation $\sigma_L$ as it is tied to (predefined) fixed step size that does not for a high precision.

## 5. Conclusion, Limitations and Outlook

We have introduced a sample-efficient approach for life testing that integrates historical data and expert knowledge. It includes a Machine Learning model and a Bayesian Inference module, which offers the possibility to derive an acquisition function. This acquisition function is easily adapted to be adversarially robust against a wrong assumption of the failure probability over stress. In a case study, we show the approach's feasibility and the acquisition function's superior behavior.

Our approach is limited by the fact that all models are equally considered in every iteration, which is disadvantageous if a model assumption proves improbable given the collected test data. In future work, we would like to enhance our approach by taking into account more potential failure models and additionally perform model selection, further lowering the number of required samples to find a good estimate of a product's expected lifetime.

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
