# OpenReview forum: "A Bayesian Approach to Adversarially Robust Life Testing"
_ICML.cc/2024/Workshop/ML4LMS — ML4LMS Poster_

### Official Review · Reviewer_mZrx · 2024-06-03
**Review for "A Bayesian Approach to Adversarially Robust Life Testing"**

**Rating:** 3
**Confidence:** 4

**Review:**

The paper presents an existing limitation in characterizing fatigue of material/structure. However, this does not seem to be an open challenge for which existing methods fail. For this reason, the paper may not gather enough interest at the workshop even if it is technically solid.